# Improved land mask for satellite remote sensing of oceans and inland waters

**Karlis Mikelsons**[1,2], and **Menghua Wang**[1]

[1]NOAA National Environmental Satellite, Data, and Information Service, Center for Satellite Applications and Research, E/RA3, 5830 University Research Court, College Park, MD 20740, USA
[2]Global Science and Technology, LLC, Greenbelt, MD 20770, USA

*Correspondence to*: Karlis Mikelsons (karlis.mikelsons@noaa.gov)

**Abstract.** We present an improved medium (250 m) spatial resolution land mask based on augmenting earlier results of Mikelsons et al. (2021) (https://doi.org/10.1016/j.rse.2021.112356) to reflect recent changes in global water surface coverage. This land mask update is critical for remote sensing of coastal oceans and inland waters as this is the first step to properly identify water pixels from land pixels for satellite data processing. We show that clear-sky false color imagery derived for monthly and yearly time periods can be effectively used to identify changes to the surface water coverage. In addition, we also use Sentinel-2 satellite imagery to derive more accurate boundaries of new water bodies with complex geometries. We demonstrate improved coverage from satellite ocean color and inland water property retrievals with the improved land mask, including a range of new inland water bodies, as well as changes to the extent of the existing water bodies. We find that majority of inland water surface changes are directly linked to human activities and list the changes to water surface areas and approximate time periods for these water bodies. The improved land mask (Mikelsons and Wang, 2025) (https://doi.org/10.17632/9r93m9s7cw.2) can also be used in the remote sensing of terrestrial, atmospheric, and cryospheric property products.

## 1. Introduction

The global surface water coverage is continuously changing. It is difficult to capture all changes as they are occurring, as some changes are very gradual, but over time quite significant. Most existing land mask datasets are static, and this has been adequate for most general and specialized applications. However, the use of static datasets necessitates periodic review of the existing land mask data for any recent changes. Furthermore, periodic review and update of existing land/water mask datasets may reveal areas where most significant, rapid, and numerous changes are taking place, which in itself is a valuable information. In addition, continuous use of the existing datasets in wider science, research, and user communities may uncover any artifacts and imprecisions that may have been previously overlooked.

One of the most significant and comprehensive efforts to map the global surface water was the landmark study by Pekel et al. (2016), which used high resolution satellite imagery and produced several metrics characterizing the global surface water (GSW), such as seasonality, occurrence, maximum extent, change, and others. However, this GSW dataset excluded polar areas, and also contained some occasional artifacts. An updated dataset was released in 2021 (https://global-surface-water.appspot.com/download) but did not mitigate some shortcomings. Nevertheless, the included metrics provide comprehensive statistical description of water surface temporal variability and ensure broad applicability for this dataset. Another land mask dataset derived from high resolution satellite measurements was created as part of an effort to map the global forest cover (GFC) (Hansen et al., 2013), yet that too excluded polar areas, and also included some occasional artifacts.

Many satellite sensors, such as the Moderate Resolution Imaging Spectroradiometer (MODIS) (Salomonson et al., 1989) on the Terra and Aqua, the Visible Infrared Imaging Radiometer Suite (VIIRS) (Goldberg et al., 2013) on the Suomi National Polar-orbiting Partnership (SNPP), NOAA-20, and NOAA-21, and the Ocean and Land Colour Instrument (OLCI) (Donlon et al., 2012) on the Sentinel-3A (S3A) and Sentinel-3B (S3B), observe earth in medium spatial resolution (~0.2–1 km), and the associated environmental research applications require a land mask dataset of a comparable spatial resolution. MODIS data were used to derive medium resolution land/water mask data (Carroll et al., 2009). The subsequent update significantly expanded locations classified as water to include occasionally and partly submerged areas (Carroll et al., 2017). The latest update (Carroll et al., 2024) appears to continue this trend and splits the MODIS-derived land mask dataset into yearly time series, but also introduces some new artifacts.

In general, the distinction between land and water surface depends on applications. This is especially true for coastal oceans and inland waters, and for higher spatial resolution data. Various types of surface water have been distinguished in numerous land cover classification studies (Brown et al., 2022; Sulla-Menashe et al., 2019; Zhang et al., 2023; Zhang et al., 2025) or within dedicated studies targeting specific water surface type (Allen and Pavelsky, 2018; Zhang and Gu, 2023), and include the temporal dynamics (Pickens et al., 2018). Land mask is especially important for satellite ocean and inland water color measurements, where it narrows down the observations to potentially valid retrievals over the water surface, and provides information about potential land adjacency effects (Bulgarelli et al., 2017). Within this context, a binary land/water mask is required to determine if satellite retrievals should be attempted over a specific geographic location. To address this need, Mikelsons et al. (2021) developed a global, medium-resolution (250 m) land mask specifically for ocean color and inland water property retrievals. They also presented a new methodology to combine multiple existing datasets to reduce artifacts and improve overall accuracy, including the high resolution GSW and GFC data and MODIS-derived medium resolution land mask data, and matching the spatial resolution of the latter. We note that, although the MODIS-derived land mask data are named as 250 m medium spatial resolution data (Carroll et al., 2017), the actual spatial resolution is closer to 230 m (7.5 arc seconds). We follow the same practice to use 250 m spatial resolution to describe our derived land mask data (actually in 7.5 arc second angular resolution in longitude and latitude).

Since then, many changes to global surface waters have taken place, many new water bodies have appeared or expanded, while others have shrunk or entirely disappeared. Occasionally, such changes can be noticed in the daily satellite imagery from polar orbiting wide-swath sensors, such as MODIS, VIIRS, and OLCI. Nevertheless, distinguishing land from water surface in daily satellite imagery is complicated due to presence of clouds (King et al., 2013), cloud shadows (Jiang and Wang, 2013), sun glint (Wang and Bailey, 2001), and occasionally heavy aerosol presence. However, changes to the water surface typically occur at more gradual seasonal or yearly time scales. Thus, representative clear-sky imagery over longer time scales may be more helpful in surface type determination. In particular, our earlier work (Mikelsons and Wang, 2021) introduced one relatively simple approach to derive clear-sky imagery from the daily multi-sensor imagery time series. This imagery, derived over an appropriate time period, eliminates the frequently changing atmospheric conditions, while retaining representative surface appearance.

In this work, we show that this clear-sky imagery can be used to identify the areas of change in global water surface, and in many cases to derive regional updates to the existing land mask. Thus, we use the previously derived land mask dataset (Mikelsons et al., 2021) and update it to incorporate the water surface changes in recent years. In this effort, we focus on new water bodies, or qualitatively significant changes, to update and improve the existing land mask dataset. While there are also more continuous and gradual changes taking place in dynamic ecosystems (such as meandering river paths, slow changes due to shifting coastlines, etc.), those have not been the main focus of this study. Although the main target use of the improved land mask dataset remains the medium resolution satellite ocean color and inland water property retrievals, we anticipate that, as before, the updated land mask dataset will have wider range of applications. The spatial resolution of the improved land mask dataset is the same as that of the earlier dataset at 7.5 arc second equal angle sampling for both longitudinal and latitudinal directions, resulting in a global dataset of 86400 × 172800 samples.

This work is structured as follows: in Section 2, we review the methodology, including use of false color imagery to derive updated land mask. In Section 3, we detail the changes and updates implemented in the new land/water mask and show improvements in the corresponding satellite ocean color and inland water property retrievals. Following the data availability statement in Section 4, we discuss the results and summarize the conclusions in Section 5.

## 2. Methodology

One of the most common satellite derived imagery types is the true color imagery, derived using the spectral bands in the red, green, and blue (RGB) parts of the visible spectrum. Satellite-measured top of the atmosphere (TOA) reflectances at each spectral band are corrected for Rayleigh scattering effects in the atmosphere (Wang, 2016), reducing the associated haze, and improving the contrast. In addition to the true color imagery, a range of other spectral band combinations are used to highlight various surface and atmospheric features. These are commonly referred to as false color imagery. One frequently used type of false color imageries is obtained by replacing the green band (typically centered around 550 nm) used for the green color channel in the imagery with the near-infrared (NIR) band (typically centered at around 865 nm) (Qi et al., 2020). This type of false color imagery is often used to distinguish surface water from land and vegetation coverage due to nearly complete water absorption at the NIR band. It is also used to identify floating algae effectively (Qi et al., 2020). In this work, we refer to it as simply "the false color imagery".

Regardless of the choice for spectral bands used in imaging, the daily satellite imagery is frequently affected by clouds and dense aerosols, preventing accurate survey of water surface extent. Furthermore, not all satellite sensors can provide complete daily

coverage. In any case, it is not practical to examine all daily satellite imagery for changes in water surface, unless automated algorithms are used. In this work, we use the clear-sky imagery derived from daily multi-sensor imagery over longer time periods to track changes to the land and water surface. The clear-sky imagery can be derived for both true and false color band combinations. For the type of false color imagery discussed here, the derived clear-sky imagery favors the overall darker water areas over lighter land (Mikelsons and Wang, 2021). Thus, clear-sky false color imagery is a proxy to maximum water extent over different time periods. We found that clear-sky true and false color imageries, which are derived over monthly and yearly time periods, are especially useful for tracking seasonal and interannual changes in water surface extent. In many frequently overcast areas, at least one month of daily imagery (sometimes more) is needed to derive clear-sky imagery. At the same time, monthly imagery can capture most seasonal changes. In comparison, yearly imagery is much easier to use, as it provides overview of the largest water extent throughout the year, but does not capture seasonal variability. The yearly imagery also can be somewhat biased towards the months with less frequent cloud coverage.

In this study, we use a combination of yearly and monthly clear-sky false color imageries from recent years (2020–2025) and compare it with the existing land/water mask to identify areas with significant changes in water coverage. For comparison, the earlier version of the land mask dataset (Mikelsons et al., 2021) was derived using a number of data sources based on satellite data from periods of 2000–2002 (Carroll et al., 2009), 2000–2015 (Carroll et al., 2017), 2000–2012 (Hansen et al., 2013), and 1984–2016 (Pekel et al., 2016). Therefore, most of them were somewhat outdated even at the time when the old land mask dataset was derived. The last dataset (Pekel et al., 2016) has since been updated to include changes up till 2021 (https://global-surface-water.appspot.com/download). We then use the monthly clear-sky false color imagery to estimate the seasonal changes for each new area found. These imagery comparisons and evaluation were conducted using the interactive features of the Ocean Color Viewer (OCView) (Mikelsons and Wang, 2018), allowing to quickly switch between the land mask and true/false color imagery, and zoom to a specific region to inspect differences at a finer detail. We note that OCView provides access to yearly and monthly global clear-sky true and false color imageries from the beginning of the VIIRS-SNPP mission in 2012. The clear-sky imagery archives from early years are derived solely from VIIRS-SNPP daily imagery. For more recent years, other available VIIRS and OLCI daily global imageries are also used to improve the accuracy of the clear-sky true color imagery.

Similarly, from 2023 onwards, the clear-sky false color imagery is derived from two VIIRS sensor daily global imageries on the SNPP and NOAA-21 satellites, including VIIRS imagery band data (Mikelsons and Wang, 2021), at the same medium spatial resolution. As such, it can be used to derive updated land/water mask in places where coastline is relatively simple and land/water reflectance contrast is high. In such cases, standard image segmentation procedures implemented in commonly available image editing software (e.g., ImageMagick, imagemagick.org) can be used to help delineate the new land and water boundaries with sufficient accuracy. However, many new water bodies have quite complicated coastlines. In these cases, we opted to use the higher spatial resolution imagery from the MultiSpectral Instrument (MSI) aboard the Sentinel-2A/B/C (Drusch et al., 2012). Following analysis of the medium resolution clear-sky imagery, we chose a representative Sentinel-2 MSI daily imagery scene clear of clouds and first derived the corresponding high resolution land mask over the region of interest using the Sentinel-2 derived true and false color imageries.

In particular, the Sentinel-2 true color imagery was derived using MSI bands 4 (665 nm), 3 (560 nm), and 2 (490 nm), for red, green, and blue channels, respectively. In false color imagery, the MSI green band was replaced by the NIR band 8 (842 nm). The spatial resolution for all Sentinel-2 MSI band used for imagery is 10 m (Drusch et al., 2012). Selected scenes of high spatial resolution Sentinel-2 true and false color imageries were passed through image segmentation procedure to produce a high resolution regional land mask for each area of interest. These high resolution regional land mask samples were then aggregated into the medium resolution land mask based on the same criteria as described in the earlier work (Mikelsons et al., 2021). Specifically, we imposed the requirement that more than 90% of high resolution (10 m) imagery pixels corresponding to the medium resolution (250 m) land mask pixel have to be identified as water in order to have the corresponding medium resolution pixel to be marked as water. We note that due to this aggregation process, the accuracy of the high resolution land mask is not crucial, since each medium resolution pixel covers more than 500 high resolution Sentinel-2 MSI derived imagery pixels. Instead, it is more important to select a representative high resolution imagery scene out of temporal time series for deriving the medium resolution land mask data.

## 3. Results

We employed the OCView web page (Mikelsons and Wang, 2018) (https://www.star.nesdis.noaa.gov/socd/mecb/color/) to survey global yearly and monthly clear-sky true and false color imageries for the most recent years (2022–2024), and compare it to the existing land mask. We note that the core functionality of OCView is described in (Mikelsons and Wang, 2018), while the process of deriving the clear-sky imagery from daily imagery time series is detailed in Mikelsons and Wang (2021). In particular, the false

color imagery is normally highly correlated to land mask, thus any changes in water surface can be easily identified by comparing these two images. Nearly all of the identified water surface changes are located in either coastal or inland areas and can be roughly divided into three types: a) changes to endorheic lakes, b) newly created inland water reservoirs due to human constructed river dams, and c) changes to coastal areas due to land reclamation or other types of developmental activities. In the following subsections, we detail each type of these changes. As a proof of utility to the improved and expanded satellite ocean/water color retrievals with the updated land mask, we also include results for chlorophyll-a (Chl-a) concentration (Hu et al., 2012; Wang and Son, 2016), and the light diffuse attenuation coefficient at 490 nm $K_d(490)$ (Wang et al., 2009).

### 3.1. Changes to endorheic lakes

In most cases, the endorheic basins have relatively simple boundaries due to water filling in relatively flat plains. In these cases, we find that deriving the land mask from the false color imagery at medium resolution is appropriate. While the spatial boundaries may not be as complex, the temporal changes can be quite frequent, often following a seasonal cycle, but also stretching over multi-year time scales. Due to ever changing nature of these water bodies, care is needed to select a representative sample for deriving the land mask.

As an example of expanded size of endorheic lakes, we show the Toshka Lakes in Egypt (Abd Ellah, 2021) (Fig. 1). These lakes are result of management in Nile's waters during recent flood events and have significantly expanded in surface area over recent years. Since these lakes have no regular inflow and outflow, they are expected to shrink unless increased precipitation in the Nile River upstream watersheds continues in the following years.

On the opposite side with shrinking size, also largely due to human activities, an example is the Aral Sea (Fig. S1 in Supplement), which has fragmented into several smaller lakes. The vanishing surface area, including causes and consequences for ecosystem changes and human activities have been subject to many studies (Shi and Wang, 2015; Wang et al., 2020). Here, we merely record the most up to date relatively stable extent of remnants of the lake as seen in clear-sky false color imagery for years 2024–2025. This represents a substantial decrease of the surface area, even compared to already diminished extent shown in the old land mask.

Other cases of endorheic lakes with changes in size and extent incorporated in the updated land mask are included in the supplementary material Section S1. All changes are summarized in Table 1. The "old" and "updated" areas listed in Table 1 refer to the areas derived from the land mask data in the earlier work (Mikelsons et al., 2021) and the current/updated version, respectively. Both of these represent estimated areas for medium (250 m) resolution satellite sensor based ocean/water color retrievals and may differ from the actual area obtained using the high resolution measurements. Most of the endorheic lakes in East Africa (primarily in Ethiopia and Tanzania) have seen increase of surface area due to increased rainfall in recent years (Byrne et al., 2024). Likewise, Lake Hulun in China has seen an expansion in recent years (Gao et al., 2024). Lastly, Lake Colhué Huapi in Argentina has disappeared for all but few weeks in the months of April and May, and has been removed in the updated water mask map.

In the context of the inland water property retrievals, it should be noted that many of the endorheic lakes tend to be very shallow and can have a high bottom reflectance. Furthermore, many are hypersaline (e.g., Aral Sea), and may have severely altered pH levels (e.g., Lake Natron), potentially complicating the efforts to retrieve water properties.

# Toshka Lakes, Egypt

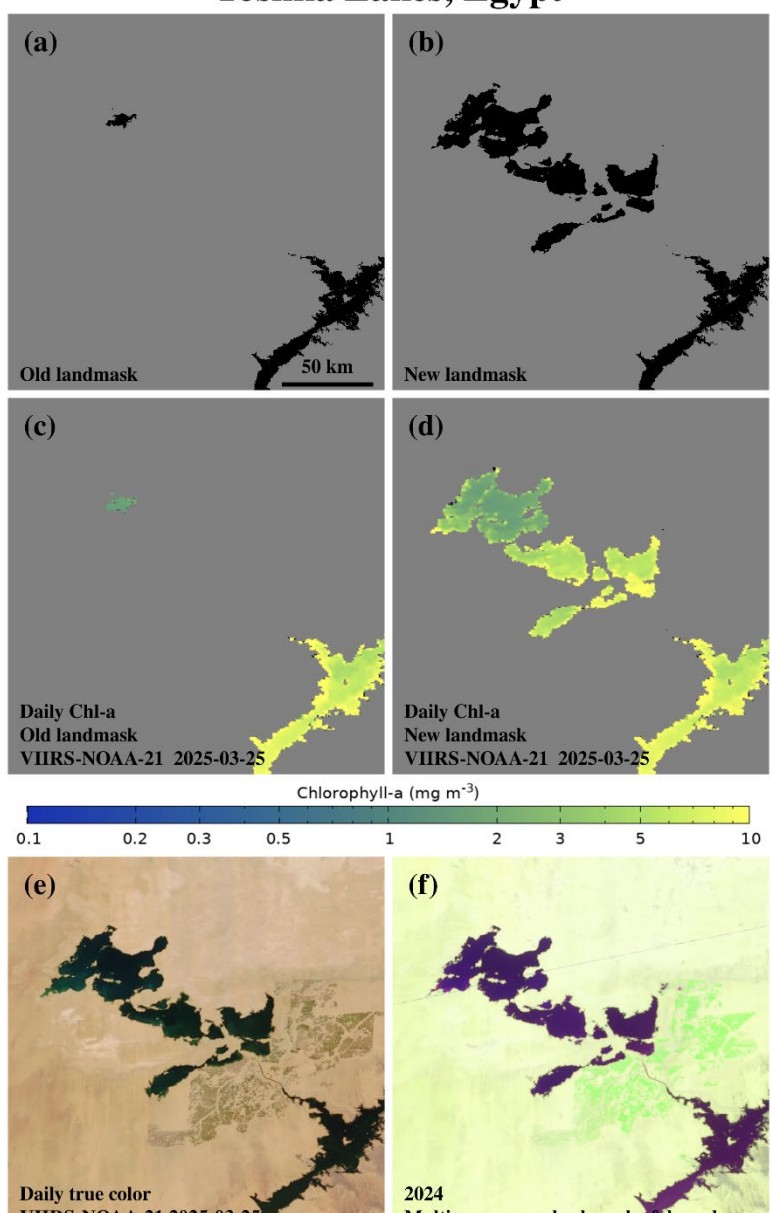

**Figure 1.** The extent of Toshka Lakes, Egypt, in the old land mask (a) and the updated land mask (b), along with the corresponding daily Chl-a retrievals [(c) and (d)] from VIIRS-NOAA-21 on March 25, 2025, and the true color imagery (e). The multi-sensor yearly clear-sky false color imagery (f) obtained from VIIRS SNPP and NOAA-21 daily false color imageries over 2024 shows nearly identical water surface coverage as the daily true color imagery (e).

**Table 1.** List of changes to endorheic lake basins in the updated land mask.

| Name and Country | Coordinates | Old Area (km$^2$) | Updated Area (km$^2$) | Figure |
|---|---|---|---|---|
| Aral Sea, Uzbekistan/Kazakhstan | ~ 45°N, 60°E | 8643 | 5010 | S1 |
| Toshka Lakes, Egypt | 23.1°N, 30.9°E | 71 | 2734 | 1 |
| Lake Abbe, Ethiopia/Djibouti | 11.15°N, 41.75°E | 150 | 414 | S2 |
| Lake Abijatta, Ethiopia | 7.6°N, 38.6°E | 62 | 160 | S3 |
| Lake Eyasi, Tanzania | 3.6°S, 35.1°E | 71 | 860 | S4 |
| Lake Manyara, Tanzania | 3.6°S, 35.8°E | 94 | 575 | S4 |
| Lake Natron, Tanzania | 2.4°S, 36.0°E | 429 | 828 | — |
| Lake Sulunga, Tanzania | 6.1°S, 35.2°E | 144 | 854 | S5 |
| Hulun Lake, China | 49.0°N, 117.5°E | 1990 | 2153 | S6 |
| Lake Colhué Huapi, Argentina | 45.5°S, 68.7°W | 253 | 0 | — |

180

185

# GERD, Ethiopia & Roseires Dam, Sudan

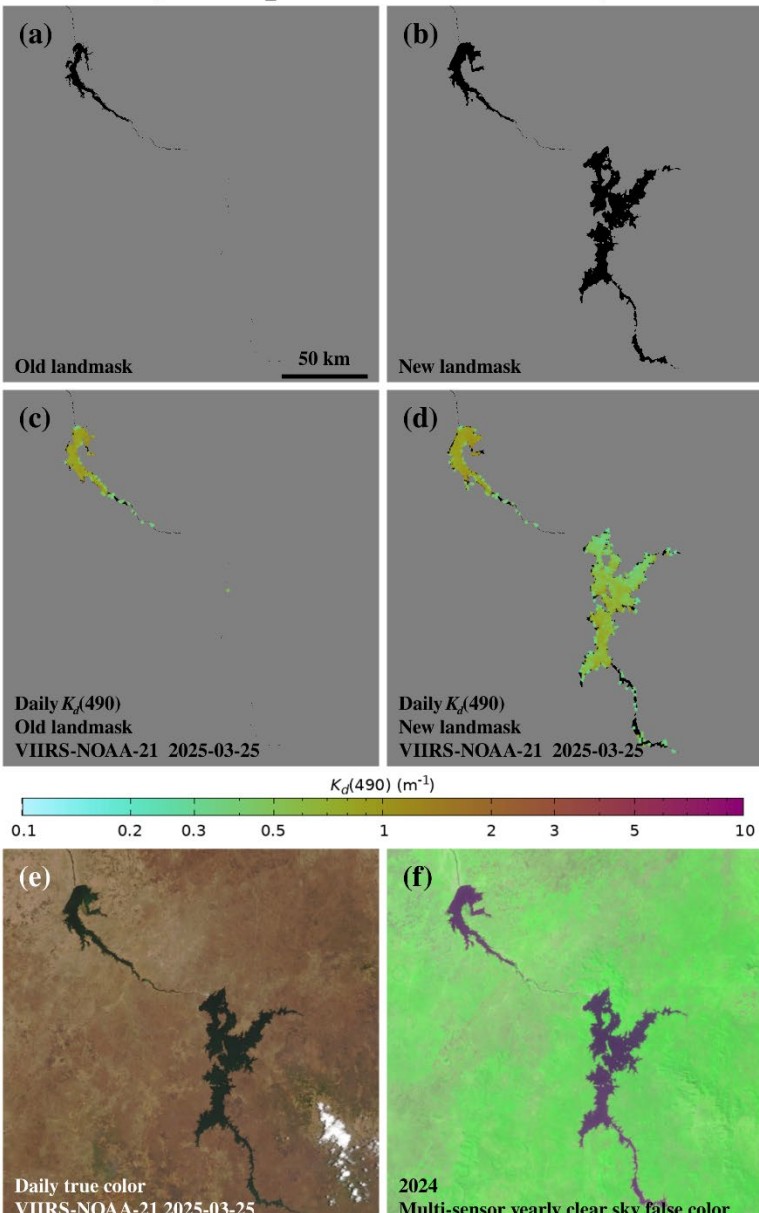

**Figure 2.** The changes in the extent of the Roseires Dam and GERD reservoirs from the old and updated land masks for (a) the old land mask with Roseires Dam reservoir in top left part, (b) the updated land mask with expanded Roseires Dam reservoir in top left and more recent GERD in the lower right, (c) VIIRS-NOAA-21-derived $K_d(490)$ using the old land mask, and (d) the same $K_d(490)$ image using the new land mask. Panel (e) is the corresponding VIIRS-NOAA-21 daily true color imagery and panel (f) is the multi-sensor yearly clear-sky false color imagery derived from VIIRS-SNPP and VIIRS-NOAA-21 daily false color imageries in 2024, showing nearly identical water surface coverage as the daily true color imagery (e).

## 3.2. New river dam impounded water reservoirs

Another major source of changes to water surface are human built dams and river filled reservoirs of water as part of hydropower and water management projects. These actions severely alter existing ecosystems and also create new habitats. Satellites provide essential measurements for understanding of changing environmental conditions, such as algae blooms in relatively static waters, sediment dynamics, etc. One of the largest recent hydro power projects is the Grand Ethiopian Renaissance Dam (GERD) (Wheeler et al., 2016; Wheeler et al., 2020) (see Fig. 2), which was absent in the old land mask. Nearby downstream Roseires dam reservoir, located across the border in Sudan, has also expanded, as compared to the extent in the old land mask, though its size still varies with seasons.

We found that most of the recently created large river dams and corresponding water reservoirs are in Africa and Asia. All new river dam water reservoirs included in the updated water mask dataset is listed in Table 2 (for Africa and South America) and Table 3 (Asia and East Europe). The river dams are listed by name, approximate geographic coordinates, the river on which each dam is constructed and that supplies water to its reservoir, the estimated new surface area, and the estimated date when the current surface extent area was reached, based on monthly clear-sky false-color imagery. We note again that the area estimate is based on suitability for medium resolution satellite inland water property retrievals, and thus will differ from the actual area measurements based on high spatial resolution imagery. In fact, in most cases the estimated surface areas for medium resolution satellite inland water property retrievals are smaller, sometimes significantly so. This is especially true for river dam water reservoirs, as these new water bodies tend to have complex, irregular shapes, which cannot be exactly represented in medium spatial resolution. Coarsening high resolution data to medium resolution data requires discarding many pixels with partial water coverage due to land contamination effects, which can severely degrade satellite water property retrievals.

We also estimate the old water surface area derived from the old water mask dataset. However, for most of newly created water bodies, the surface area in the old land mask data is either zero, or very small (<10%), as compared to the updated one. Only two of the water reservoirs listed in Table 2 are older and recently expanded to larger surface area. The Roseires Dam reservoir in Sudan has seen recent expansion (though with substantial seasonal variability) from 226 km$^2$ in the old land mask to 332 km$^2$ in the updated land mask, representing almost 50% increase. The reservoir created by Mtera Dam in Tanzania, though over 40 years since its completion, also has expanded in recent years and this change is reflected in the updated water mask as an increase of surface area from 254 km$^2$ to 577 km$^2$. We also found two recent river dam reservoirs in South America, both on Teles Pires River in Brazil (also listed in Table 2). We did not find significant changes to water surface extent of any type in the Central and North America.

Globally, the largest number of new river dam water reservoirs were found in Asia, listed in Table 3. As one of examples for relatively recent water bodies with a fairly complex shape, we highlight reservoir impounded by the Lower Se San Dam 2 in Cambodia in Fig. 3 (Sithirith, 2021). Here, and in similar cases, we used representative high resolution Sentinel-2 MSI imagery to derive the medium resolution land mask. Nevertheless, all of the new water bodies were first identified in the medium resolution clear-sky false color imagery.

Again, as seen in the results for Africa, for most of the new river reservoir based water bodies, the area in the old water mask is relatively small (<10%), as compared to the area estimated from the updated land mask dataset. The only significant exception is the reservoir bounded by Sriram Sagar Dam in India, which was completed in 1977. While this reservoir has existed for decades, we estimate that it has increased in size from 91 km$^2$ in the old land mask to 176 km$^2$ in the updated dataset. Table 3 also lists the only river fed water body of decreased size – the collapse of reservoir on Dnipro river as a result of destruction of Kakhovka Dam in Ukraine (Vyshnevskyi and Shevchuk, 2024), which had the estimated water surface area of 2017 km$^2$ in the old water mask and now is reduced to just 69 km$^2$ in the updated one.

**Table 2.** List of new and changed river dam reservoirs in Africa and South America.

| Name and Country | Coordinates | Estimated New Area (km$^2$) | River | Recent changes | Figure |
|---|---|---|---|---|---|
| Grand Ethiopian Renaissance Dam, Ethiopia | 11.21°N, 35.09°E | 1298 | Blue Nile | 2024-12 | 2 |
| Roseires Dam, Sudan | 11.80°N, 34.39°E | 332 | Blue Nile | 2024-11 | 2 |
| Genale Dawa III Power Station, Ethiopia | 5.61°N, 39.69°E | 76 | Ganale Doria | 2019-12 | S7 |
| Mtera Dam, Tanzania | 7.14°S, 35.98°E | 577 | Great Ruaha | 2024-03 | S5 |
| Julius Nyerere HPS, Tanzania | 7.80°S, 37.83°E | 681 | Rufiji | 2024-03 | S8 |
| Calueque Dam, Angola | 17.27°S, 14.55°E | 78 | Cunene | 2024-01 | S9 |
| Laúca Dam, Angola | 9.74°S, 15.13°E | 168 | Cuanza | 2018-05 | S10 |
| Lom Pangar Dam, Cameroon | 5.38°N, 13.5°E | 182 | Lom | 2016-11 | S11 |
| Kashimbila Dam, Nigeria | 6.87°N, 9.76°E | 36 | Katsina Ala | 2017-11 | S12 |
| Zungeru Dam, Nigeria | 9.90°N, 6.30°E | 334 | Kaduna | 2021-12 | S13 |
| Unnamed Dam, Burkina Faso | 13.36°N, 2.05°W | 19 | White Volta | 2017-10 | S14 |
| Samendéni Dam, Burkina Faso | 11.38°N, 4.58°W | 86 | Black Volta | 2018-10 | S15 |
| Souapiti Dam, Guinea | 10.42°N, 13.25°W | 91 | Konkouré | 2021-12 | S16 |
| Colíder Dam, Brazil | 10.98°S, 55.77°W | 116 | Teles Pires | 2018-04 | S17 |
| Sinop Dam, Brazil | 11. 27°S, 55.45°W | 142 | Teles Pires | 2019-05 | S17 |

**Lower Se San 2 Dam, Cambodia**

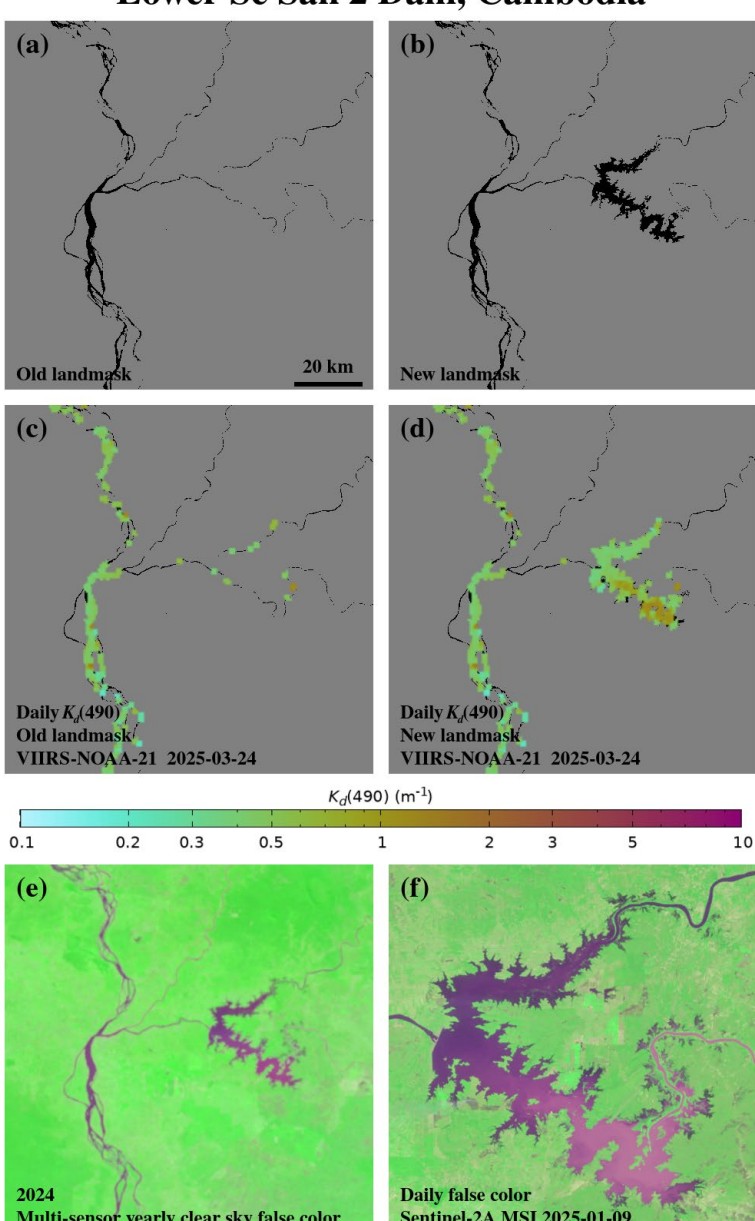

**Figure 3.** The land mask changes in lower Mekong River basin in Cambodia for (a) the old land mask showing Mekong River and its tributaries in the northern part of Cambodia, (b) updated land mask with the reservoir created by Lower Se San Dam 2 (right side), and VIIRS-NOAA-21-derived daily $K_d(490)$ with (c) the old and (d) the updated land mask, respectively. Panel (e) shows the corresponding yearly clear-sky false color imagery and panel (f) shows the detailed view of the impounded basin in the daily false color imagery derived from Sentinel-2A MSI scene, captured on 9 January 2025.

**Table 3.** List of new and changed river dam reservoirs in Asia and East Europe.

| Name and Country | Coordinates | Estimated new area (km²) | River | Recent changes | Figure |
|---|---|---|---|---|---|
| Nizhne-Bureyskaya Dam, Russia | 49.79°N, 129.98°E | 123 | Bureya | 2019-05 | S18 |
| Baihetan Dam, China | 27.22°N, 102.90°E | 135 | Jinsha | 2021-09 | S19 |
| Wendegen Reservoir, China | 46.9°N, 121.94°E | 75 | Chuoer | 2024-08 | S20 |
| Geshan Dam, China | 47.36°N, 127.49°E | 27 | Nuomin | 2022-09 | S21 |
| Pubugou Dam, China | 29.21°N, 102.83°E | 58 | Dadu | before 2012 | S22 |
| Chushuidian Dam, China | 32.25°N, 113.96°E | 24 | Huaihe | 2020-09 | S23 |
| Ban Pook Dam, Laos | 16.36°N, 106.24°E | 21 | — | 2019-10 | S24 |
| Nam Theun 1 HPP, Laos | 18.36°N, 104.15°E | 34 | Nam Kading | 2022-10 | S25 |

240

| Nam Ngiap 1 Dam, Laos | 18.65°N, 103.52°E | 34 | Nam Ngiap | 2020-01 | S26 |
|---|---|---|---|---|---|
| Nam Khong 1 Dam, Laos | 14.55°N, 106.74°E | 14 | Nam Khong | 2021-11 | S27 |
| Nam Khong 2 Dam, Laos | 14.50°N, 106.86°E | 2 | Nam Khong | 2022-11 | S27 |
| Nam Khong 3 Dam, Laos | 14.57°N, 106.92°E | 12 | Nam Khong | 2022-11 | S27 |
| Xe Namnoy Dam, Laos | 15.03°N, 106.6°E | 17 | Xe Namnoy | 2020-11 | S27 |
| Lower Se San 2 Dam, Cambodia | 13.55°N, 106.26°E | 177 | Tonlé San | 2018-10 | 3 |
| Prakaet Dam, Thailand | 13.09°N, 101.82°E | 5 | Prakaet | 2018-11 | S28 |
| Hang Maeo Dam, Thailand | 13.07°N, 101.97°E | 9 | Hang Maeo | 2023-11 | S28 |
| Jatigede Dam, Indonesia | 6.86°S, 108.10°E | 28 | Manuk | 2016-06 | S29 |
| Myittha Dam, Myanmar | 21.99°N, 94.04°E | 14 | Myittha | 2016-10 | S30 |
| Hiramandalam Dam, India | 18.67°N, 83.93°E | 8 | Minor stream | 2018-11 | S31 |
| Kundaliya Dam, India | 23.92°N, 76.31°E | 23 | Kali Sindh | 2018-09 | S32 |
| Mohanpura Dam, India | 23.96°N, 76.78°E | 35 | Newaj | 2018-09 | S32 |
| Lower Indra Dam, India | 20.39°N, 82.67°E | 21 | Indra | 2018-09 | S33 |
| Machagora Dam, India | 22.12°N, 79.16°E | 22 | Pench | 2016-10 | S34 |
| Mallana Sagar Dam, India | 17.96°N, 78.74°E | 25 | Minor stream | 2021-12 | S35 |
| Mid Manair Dam, India | 18.39°N, 78.96°E | 46 | Manair | 2019-11 | S35 |
| Sriram Sagar Dam, India | 18.96°N, 78.34°E | 176 | Godavari | before 2012 | S36 |
| Moragahakanda Dam, Sri Lanka | 7.70°N, 80.77°E | 18 | Amban Ganga | 2018-01 | S37 |
| Kalu Ganga Dam, Sri Lanka | 7.56°N, 80.83°E | 5 | Kalu Ganga | 2020-01 | S37 |
| Yan Oya Dam, Sri Lanka | 8.74°N, 80.88°E | 34 | Yan Oya | 2019-02 | S38 |
| Ilısu Dam, Turkey | 37.53°N, 41.85°E | 72 | Tigris | 2020-05 | S39 |
| Alpaslan-2 Dam, Turkey | 39.04°N, 41.52°E | 43 | Murat | 2021-05 | S40 |
| Kakhovka Dam, Ukraine | 46.78°N, 33.37°E | 69 | Dnipro | 2023-07 | S41 |

### 3.3. Changes to coastal regions

Another type of human induced changes to global surface waters is due to land reclamation, and changes to land and water surface cover in coastal regions. Such changes are typically very gradual and can be readily identified in satellite daily imagery or clear-sky imagery derived over a longer time period. Figure 4 shows such changes near the coastline of the United Arab Emirates (UAE), with the updated land mask (Fig. 4b) closely resembling the true and false color imagery (Fig. 4c and 4d, respectively). The updated land mask (Fig. 4b) adds offshore Crescent Island and newly reclaimed lands near the UAE coastline (Subraelu et al., 2022), absent in the old land mask (Fig. 4a). Other recorded areas of land reclamation projects include establishment of new polders in Markermeer, Netherlands, and a port expansion in Singapore. However, the largest total area of changes between the old and updated land masks were seen in Egypt, partly due to expansion of the Suez Canal, but mostly due to changes in the nearby Nile wetlands, also related to human activities. This includes both areas previously seen as water in the old land mask and identified as land in the updated one, and vice versa.

**UAE coastline**

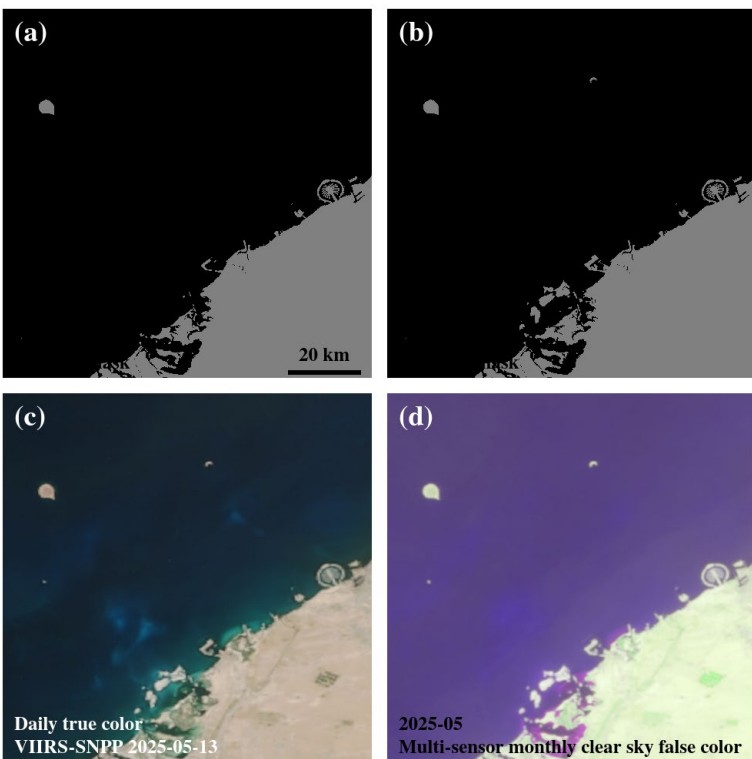

**Figure 4.** Changes between the old and updated land masks due to land reclamation projects near the UAE coastline of (a) old land mask before changes, (b) the updated land mask showing the Crescent Island (top center right) and large scale land reclamation near coast, (c) VIIRS-SNPP true color imagery on May 13, 2025, and (d) monthly clear-sky imagery derived from daily VIIRS-SNPP and VIIRS-NOAA-21 false color imageries in May 2025.

**Table 4.** List of changes to coastal areas.

| Name and Country | Coordinates | Estimated area of changes (km$^2$) | Type of changes | Figure |
|---|---|---|---|---|
| Port Said and Suez Canal, Egypt | 7.13°S, 35.98°E | 453 | Artificial lake and waterway development | S44 |
| Marker Wadden, Netherlands | 52.59°N, 5.38°E | 8 | Land reclamation | S43 |
| Trintelzand, Netherlands | 52.65°N, 5.38°E | 6 | Land reclamation | S43 |
| Strandeiland, Netherlands | 52.36°N, 5.02°E | 7 | Land reclamation | S43 |
| Tuas, Singapore | 1.23°N, 103.63°E | 22 | Land reclamation | S42 |
| Crescent Island, UAE | 25.31°N, 54.65°E | 1 | Land reclamation | 4 |
| UAE coastline | 24.75°N, 54.56°E | 52 | Land reclamation | 4 |

### 3.4. Fixing artifacts in the earlier land mask dataset

Lastly, we have fixed a couple of artifacts found in the earlier version of the water mask dataset. One of these was found in the Arctic near East Greenland. Since not all data sources in our earlier study covered the polar regions, fewer data sources were used, and the results were more prone to errors. In particular, two of the MODIS-derived data sources, MOD44Wv5 land mask (Carroll et al., 2009) and MOD44Wv6 land mask (Carroll et al., 2017), had an outsized weight, and that caused their artifacts to propagate into the derived water mask data. In this updated version, we have corrected this artifact by using the OpenStreetMap (https://openstreetmap.org) data, which we also find as consistent with recent yearly clear-sky true and false color imageries.

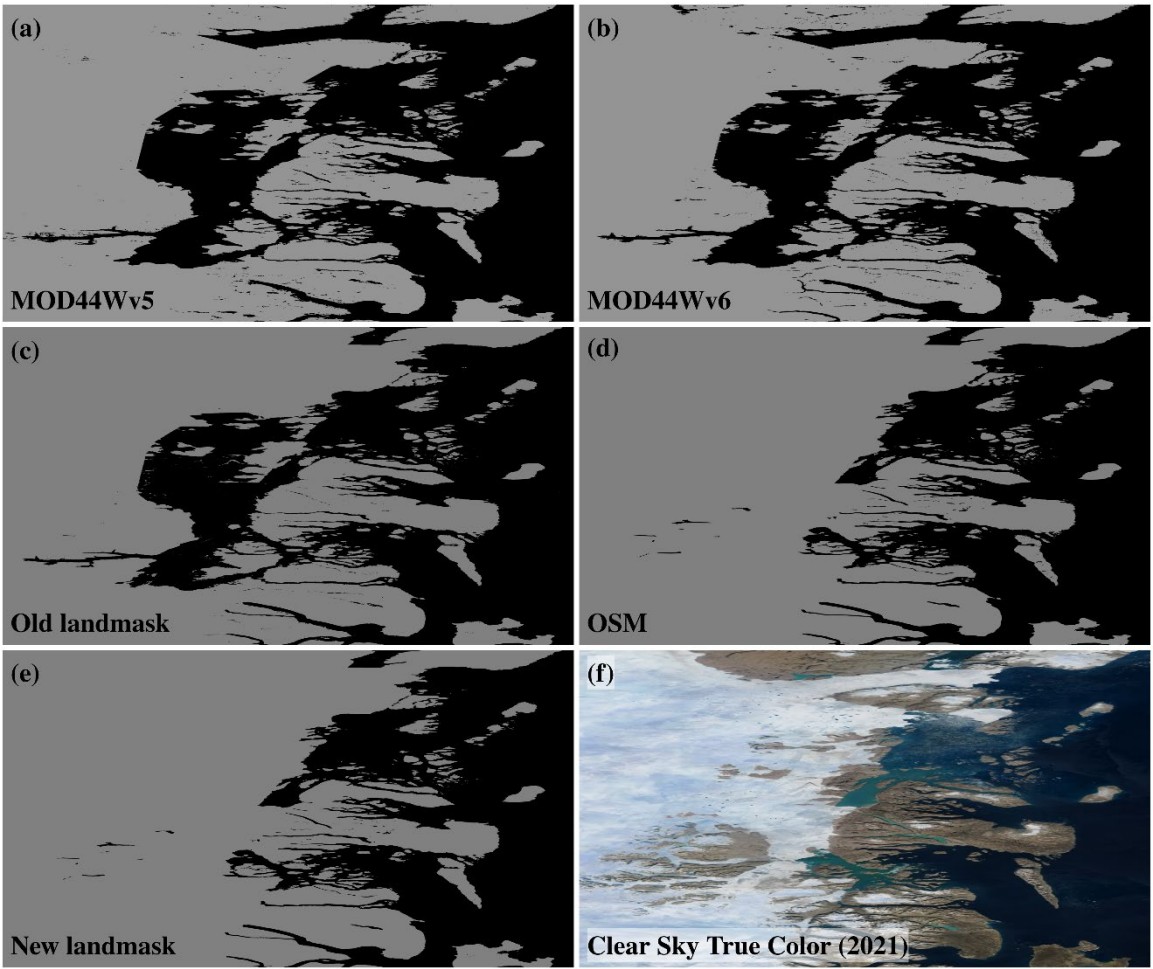

Figure 5. Area of East Greenland showing artifacts in MOD44Wv5 (a) and MOD44Wv6 (b) land masks used as sources for producing the old land mask (c) and propagating artifacts to it. In contrast, OSM data (d) are much more consistent with the yearly multi-sensor clear-sky true color imagery (f) obtained from the daily VIIRS and OLCI sensor imagery in 2024 and were used as data source to produce the updated land mask (e) over this area.

We also fixed another artifact and improved the land mask dataset to include recent changes in the Amazon River Delta region. Here, deriving accurate land mask dataset is challenging due to a number of factors. Frequently cloudy skies limit the number of usable satellite observations in the region, and relatively high tidal amplitudes cause rapid diurnal changes in large tidal areas. In addition, high sediment loads in waters elevate the reflectance in the NIR part of spectrum and make it more difficult to distinguish the sediment rich waters from the adjacent wetlands in the false color imagery. These factors were the likely causes of the artifacts in the sources used to derive the old land mask dataset. The GFC dataset, in particular, misclassified large areas of shallow sediment rich waters north of the Amazon River Delta as land, and GSF data were also somewhat affected. The artifacts from these sources propagated to the old land mask dataset (Fig. S45a) and have been corrected in the new land mask data (Fig. S45b). Furthermore, we also identified many natural changes to water surface extent in the Amazon River Delta region, including changes to river paths and shifting coastlines, and incorporated them in the new land mask dataset. Since we could not find a single recent cloud free Sentinel-2 MSI scene, we opted to use a number of (around 50) recent (2024–2025) Sentinel-2 MSI scenes over this region to derive clear-sky true and false color imageries in high spatial resolution. This false color imagery was then used to derive an updated medium resolution land mask using the same methodology as described above. The resultant land mask roughly represents the extent of land at a medium tidal water height.

## 4. Data availability

Both the earlier and the updated land/water mask data are publicly available (Mikelsons and Wang, 2025, https://doi.org/10.17632/9r93m9s7cw.2). Interactive visualization of the updated land mask data is also available on the OCView

website (https://www.star.nesdis.noaa.gov/socd/mecb/color/ocview/ocview.html), along with the monthly global clear-sky true and false color imagery used in this study.

## 5. Discussion and conclusions

We have derived an updated global medium (250 m) resolution land mask, incorporating the changes to the global water surface over the past decade. In particular, we have also shown that clear-sky imagery derived from multi-sensor daily imagery time series can be a valuable resource to evaluate the accuracy of the existing land mask datasets, and to identify recent changes in the global water surface.

We find that most common water surface changes are due to human activities, such as newly constructed river dams, or land
reclamation projects in coastal regions. Changes to the endorheic lakes found in arid regions also can be due to human water use, but in most cases, these are driven by interannual changes to the upstream rainfall amounts.

While the target application for this updated land mask dataset is medium resolution satellite ocean/water color measurements, we expect it to be useful in other types of remote sensing applications. In fact, we argue that new water bodies often display the most rapid environmental changes, and are of particular interest to the research community, including remote sensing of terrestrial,
atmospheric, and cryospheric properties. We also note that for satellite sensors with long mission lifespans (currently VIIRS-SNPP, MODIS-Aqua, and at some point, OLCI-Sentinel-3A), different land mask datasets are required to accurately represent the various time periods of the mission.

While this study presents a static binary global land mask dataset, it is clear that, in the regions where water coverage follows a clear seasonal cycle, satellite ocean/water color retrievals would benefit from a seasonally resolved land mask dataset. However,
this will require much more efforts with detailed validations. Likewise, inclusion of water fractional coverage data in land mask dataset may also be beneficial for many applications, such as more accurate evaluation of coastal adjacency effects in future satellite water color retrieval algorithms.

Overall, periodic updates to the global land mask data are essential to maintain accuracy. In fact, we can gauge the temporal frequency of changes to global water surface by looking at the estimated time of the most recent changes (as well as quantitative
variations to the water surface area) listed in Tables 2 and 3. From these data, we suggest that, at a global scale, about 3–5 year update cycle may be adequate for medium resolution land mask datasets used in satellite ocean/water color studies. Local, regional, and high spatial resolution land mask datasets likely require more frequent updates. Ultimately, the update frequency is also affected by practical considerations, such as available research time and resources.

In documenting the latest significant regional changes to the global land mask dataset, we also acknowledge that incremental
updates such as those detailed in this study may be time consuming and potentially prone to some form of human bias. In contrast, automated methods have been widely used for mapping global water and land cover extents and require less human involvement. We surmise that the process of extracting the land mask from the clear-sky imagery employed in this work may also be automated, using an existing or a custom-tailored approach. Nevertheless, as seen in this and our earlier work, automated approaches can also lead to artifacts. Thus, careful validation is always necessary, especially for global studies covering a wide range of land surface
types and water optical properties. In fact, having surveyed and evaluated a number of data sources derived by a variety of mostly automated approaches, we would like to stress the importance of data validation by a human expert. In this task, we found that interactive visualization platforms (such as OCView) are immensely useful, including the capability to inspect imagery at different spatial resolutions, and to quickly switch and compare different data sources and products, all while maintaining the geographical context. Use of automated methods is likely the only viable option for working with and deriving of global high spatial resolution
land mask datasets due to large data volumes, and the use of interactive multi-resolution imagery in validation can be especially useful. The Sentinel-2 MSI satellite series is seen as one of the leading high resolution environmental data sources with good spatial and spectral resolutions, and data are publicly available. Use of commercially sourced satellite data imagery may help to increase the observation frequency, but may be cost prohibitive for global applications.

## Author contributions

KM: Conceptualization, methodology, software, analysis, validation, visualization, writing – original draft preparation, writing – review and editing.

MW: Conceptualization, project administration, supervision, funding acquisition, writing – review and editing

**Competing interests**

The authors declare that they have no conflict of interest.

**6. Acknowledgments**

This work was supported by the Joint Polar Satellite System (JPSS) funding. We thank two anonymous reviewers for their useful comments. We also thank Dr. Jackson Tan (NASA GSFC) for pointing out an artifact in the earlier land mask data (addressed in Section 3.4). Sentinel-2 MSI Level-1B data used in this study were downloaded from the Copernicus Data Space Ecosystem
Browser    (https://browser.dataspace.copernicus.eu).    Scientific    color    maps    "Hawaii"    and    "Imola" (https://doi.org/10.5281/zenodo.1243862) were used to plot ocean color data in figures. The work was supported by the Joint Polar Satellite System (JPSS) funding. Part of this work was performed and funded under ST133017CQ0050_1332KP22FNEED0042. The scientific results and conclusions, as well as any views or opinions expressed herein, are those of the author(s) and do not necessarily reflect those of NOAA or the Department of Commerce.

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
