# Peer review of "Improved land mask for satellite remote sensing of oceans and inland waters"

_Earth System Science Data, 2025_

## Author Response (AR1)

**RC1**: ['Comment on essd-2025-543'](), Anonymous Referee #1, 13 Oct 2025

**Citation**: https://doi.org/10.5194/essd-2025-543-RC1

Overall, this kind of data which updated to recent state of water area is useful for the processing especially for algorithms dedicated to the ocean area. The one-file and one-byte format is convenient to use in processing data from the low-middle resolution global observation sensors. However the seasonal cycle may also affect the OC processing in the coast of the lake or big rivers and need to be considered. This data is indeed valuable in the current state, however it may have more value if it includes information of seasonal cycle orr probability (or coverage) of water cover within the pixels which may be obtained by authors already.

I think this datasets has the publication quality but it may be better to consider the following comments.

Author response:

We thank the Reviewer #1 for a positive review and relevant questions.

We agree that a land mask with seasonal variability would be more accurate for coastal and inland waters. This work represents an incremental improvement over the earlier version of the land mask, which did not include the seasonal cycle. Thus, incorporating the seasonal cycle would require deriving an entirely new data set, with changes to methodology, and much more careful and detailed validation, that may perhaps be a topic for a future work. This work addresses the most immediate needs for satellite ocean/water color retrievals, including recently established permanent inland water bodies and removing the obvious artifacts in the earlier version.

Likewise, we agree that the water pixel fractional coverage may be useful in many applications. In fact, some (but not all) of the data sources used to derive the previous/original version of this land mask data set were of higher spatial resolution ($\sim$ 30 m), allowing to estimate the fractional water coverage. However, a consistent global data set for fractional water coverage would require additional independent high resolution data sources, and more thorough validation. In satellite ocean/water color retrievals, there is a clear need for a binary land/water data set to determine if water color retrievals should be attempted. For these reasons, we opted to keep the same binary land mask format in this work. We have included a paragraph in the discussion to highlight these potential future improvements:

"While this study presents a static binary global land mask data set, it is clear that in the regions where water coverage follows a clear seasonal cycle, satellite ocean/water color retrievals would benefit from a seasonally resolved land mask data set. However, this will require much more efforts with detailed validations. Likewise, inclusion of water fractional coverage data in land mask data set may also be beneficial for many applications, such as more accurate evaluation of coastal adjacency effects in future satellite water color retrieval algorithms."

Specific comments:

Line-100 "In this study, we use a combination of yearly and monthly clear sky false color imageries from the recent years (2020–2025) ..";

Do you have confirmed that the water areas have not changed except for areas listed in tables in this paper?

Author response:

We have attempted to capture all the significant recent changes in land/water coverage in this updated data set. We did not focus on capturing all minor changes (some of which may not have a great effect in the medium spatial resolution), which we also have stated in the introduction (lines 67 - 70). We did perform a thorough evaluation of the previous and the updated land masks using the available clear sky false color imagery for the recent years.

Line-123 " we imposed the requirement that more than 90% of high resolution (10 m) imagery pixels":

Can you clarify that the 90% was applied to only spatial coverage or both spatial and temporal (considering seasonal change) coverage?

Author response:

That is correct, the 90% threshold was applied only to the spatial coverage, not the temporal. We have added a sentence to clarify this:

"Instead, it is more important to select a representative high resolution imagery scene out of temporal time series for deriving the medium resolution data."

Line-295 "Overall, periodic updates are essential to maintain accuracy.":

Can you propose the update period recommended for the natural and artificial changes?

Author response:

We have added a paragraph in the Discussion and Conclusions section to address this question:

"Overall, periodic updates to the global land mask data are essential to maintain accuracy. In fact, we can gauge the temporal frequency of changes to global water surface by looking at the estimated time of the most recent changes (as well as quantitative variations to the water surface area) listed in Tables 2 and 3. From these data, we suggest that, at a global scale, about 3 – 5 year update cycle may be adequate for medium resolution land mask data sets used in satellite ocean/water color studies. Local, regional, and high spatial resolution land mask data sets likely require more frequent updates. Ultimately, the update frequency is also affected by practical considerations, such as available research time and resources. "

**RC2**: ['Comment on essd-2025-543'](), Anonymous Referee #2, 10 Nov 2025

**Citation**: https://doi.org/10.5194/essd-2025-543-RC2

SUMMARY

This manuscript described an updated global land-ocean mask representative of the period 2020 to 2025 and contrasted it with the previous mask to identify secular changes due to water management practices. The new mask refined the mask by incorporating additional satellite information, fixed artifacts in the previous mask, and captured changes of water extent. On the latter subject, the authors detailed the contraction and expansion of several lakes due to water management practices or shifts in rainfall pattern, the emergence of lakes due to new dams, and changes in coastlines due to activities such as land reclamation.

This is a clear, well-written manuscript, not only describing the improvement to an existing product, but also catalogued how human activities are impacting the coverage of water surfaces and the importance of capturing these impacts. The data is readily accessible and appears to be of high quality. There are a few minor points that can help refine the manuscript, but this is a study worthy of publication in this journal.

Author response:

We thank the Reviewer #2 for a positive response and relevant comments and suggestions on how to improve our manuscript.

COMMENTS

1) Title/abstract: Both the title and abstract convey the impression that this manuscript describes a new product, but a large part of it in fact assesses changes in the water extent due to human activities. I consider this a plus, since it goes beyond the simplistic "here's a new product" to connect the product to the real world. But this should be mentioned in the abstract at the minimum, even though I recognize that this journal focuses on the data rather than results.

Author response:

We have included another sentence in the abstract to state that most of the changes to global water surface extent are directly linked to human activities:

"We find that majority of inland water surface changes are directly linked to human activities and list the changes to water surface areas and approximate time periods for these water bodies."

2) L52-55: I may have missed this elsewhere in the manuscript, but is the product on an "equal-area" grid at 230 m or a typical "equal-angle" grid (i.e., the standard rectangular latitude-longitude grid) that has a resolution of 230 m at the equator (but not at other latitudes)? I suspect it is the latter, in which case please mention the shape of the global array (e.g., "360 × 180" for a 1° grid).

Author response:

The reviewer is correct that the land mask data set is sampled on an "equal angle" grid, with grid of size 86400 x 172800, or 7.5 arc seconds of longitude/latitude per sample. We have made changes to clarify this:

Line 56: "… (actually in 7.5 arc second angular resolution in longitude and latitude)."

Lines 73 - 75: "The spatial resolution of the improved land mask data set is the same as that of the earlier data set at 7.5 arc second equal angle sampling for both longitudinal and latitudinal directions, resulting in a global data set of 86400 × 172800 samples."

3) L100-101: For context, please provide the approximate period that the existing mask represents.

Author response:

We have added a sentence in the manuscript to clarify this:

"For comparison, the earlier version of the land mask data set (Mikelsons et al., 2021) was derived using a number of data sources based on data from periods of 2000–2002 (Carroll et al., 2009), 2000–2015 (Carroll et al., 2017), 2000–2012 (Hansen et al., 2013), and 1984–2016 (Pekel et al., 2016). Therefore, most of them were somewhat outdated even at the time when the old land mask data set was derived. The last dataset (Pekel et al., 2016) has since been updated to include changes up till 2021 (https://global-surface-water.appspot.com/download)."

4) L129-131: Looking at the website, the tool is labeled as "experimental", suggesting that it may not be permanent. Therefore, the authors should provide a description of OCView to a degree such that a researcher could reasonably replicate its functions should it no longer exist in the future.

Author response:

We have added a sentence to provide published references to the tools and methods used in this work, which should offer sufficient details on how to reproduce the results (lines 139 -141):

"We note that the core functionality of the OCView is described in (Mikelsons and Wang, 2018), while the process of deriving the clear sky imagery from daily imagery time series is detailed in (Mikelsons and Wang, 2021)."

5) L295-298: These are critical points: the entire manuscript underscores the importance of capturing changes in water extent but also the challenges in doing so. I would like to see a speculative discussion, especially from the perspectives of the authors who have done the actual hard work, on the feasibility of automation (e.g., can the clear sky imagery approach be automated?), periodic updates (e.g., how often is reasonable?), and the use of commercial satellite imagery. Such information would be highly beneficial for anyone in the community considering such an endeavor, though I would also understand if the authors prefer to exercise some restrain over such speculations.

Author response:

We thank the reviewer for appreciating the difficulty of the task. We have expanded the Discussion and Conclusions section to share our perspective on these questions (lines 311 – 331):

"Overall, periodic updates to the global land mask data are essential to maintain accuracy. In fact, we can gauge the temporal frequency of changes to global water surface by looking at the estimated time of the most recent changes (as well as quantitative variations to the water surface area) listed in Tables 2 and 3. From these data, we suggest that, at a global scale, about 3 – 5 year update cycle may be adequate for medium resolution land mask data sets used in satellite ocean/water color studies. Local, regional, and high spatial resolution land mask data sets likely require more frequent updates. Ultimately, the update frequency is also affected by practical considerations, such as available research time and resources.

In documenting the latest significant regional changes to the global land mask data set, we also acknowledge that incremental updates such as those detailed in this study may be time consuming and potentially prone to some form of human biases. In contrast, automated methods have been widely used for mapping global water and land cover extents and require less human involvement. We surmise that the process of extracting the land mask from the clear sky imagery employed in this work may also be automated, using an existing or a custom-tailored approach. Nevertheless, as seen in this and our earlier work, automated approaches can also lead to artifacts. Thus, careful validation is always necessary, especially for global studies covering a wide range of land surface types and water optical properties. In fact, having surveyed and evaluated a number of data sources derived by a variety of mostly automated approaches, we would like to stress the importance of data validation by a human expert. In this task, we found that interactive visualization platforms (such as OCView) are immensely useful, including the capability to inspect imagery at different spatial resolutions, and to quickly switch and compare different data sources and products, all while maintaining the geographical context. Use of automated methods is likely the only viable option for work with and deriving of global high spatial resolution land mask data sets due to large data volumes, and the use of interactive multi resolution imagery in validation can be especially useful here. The Sentinel-2 MSI satellite series is seen as one of the leading high resolution environmental data sources with good spatial and spectral resolutions, and data are publicly available. Use of commercially sourced satellite data imagery may help to increase the observation frequency but may be cost prohibitive for global applications."

**Additional changes:**

In the revised manuscript, we also made following additional changes:

- Added few missing references
- Removed repetitive "Fig." from the last columns in Tables 1 – 4, leaving only the figure numbers
- Replaced "$K_d$" with italicized "$K_d$" in the labels on Figures 2 and 3 in the manuscript, and the relevant Figures in Supplement
- Added acknowledgement to JPSS funding and to the reviewers for their comments
- Updated first author's second affiliation from "Global Science and Technology, Inc" to "Global Science and Technology, LLC"
- Made additional minor formatting changes and language/grammar corrections

---

## Author Response (AR2)

Dear Editor(s),

In this second revision, we have made further minor changes to correct few language related issues and added one missing reference. We would like to see if this version can be accepted as the final before our submission moves on to the copyediting stage.

Thank you,

Karlis Mikelsons and Menghua Wang

---

## Author Response (AR3)

Dear Editor(s),

In this third revision, we have made further minor formatting changes and language/grammar corrections, and added one missing reference. We also fixed incorrectly swapped panel labels (c) and (d) in Figure 5.

Thank you,

Karlis Mikelsons and Menghua Wang